# Learning Efficient Algorithms
# with Hierarchical Attentive Memory

**Marcin Andrychowicz**[*]
Google Deepmind

**Karol Kurach**[*]
Google / University of Warsaw

## Abstract

In this paper, we propose and investigate a novel memory architecture for neural networks called Hierarchical Attentive Memory (HAM). It is based on a binary tree with leaves corresponding to memory cells. This allows HAM to perform memory access in $\Theta(\log n)$ complexity, which is a significant improvement over the standard attention mechanism that requires $\Theta(n)$ operations, where $n$ is the size of the memory. We show that an LSTM network augmented with HAM can learn algorithms for problems like merging, sorting or binary searching from pure input-output examples. In particular, it learns to sort $n$ numbers in time $\Theta(n \log n)$ and generalizes well to input sequences much longer than the ones seen during the training. We also show that HAM can be trained to act like classic data structures: a stack, a FIFO queue and a priority queue.

## 1 Intro

Deep Recurrent Neural Networks (RNNs) have recently proven to be very successful in real-word tasks, e.g. machine translation (Sutskever et al., 2014) and computer vision (Vinyals et al., 2014). However, the success has been achieved only on tasks which do not require a large memory to solve the problem, e.g. we can translate sentences using RNNs, but we cannot produce reasonable translations of really long pieces of text, like books.

A high-capacity memory is a crucial component necessary to deal with large-scale problems that contain plenty of long-range dependencies. Currently used RNNs do not scale well to larger memories, e.g. the number of parameters in an LSTM (Hochreiter & Schmidhuber, 1997) grows quadratically with the size of the network's memory. In practice, this limits the number of used memory cells to few thousands.

It would be desirable for the size of the memory to be independent of the number of model parameters. The first versatile and highly successful architecture with this property was Neural Turing Machine (NTM) proposed by Graves et al. (2014). The main idea behind the NTM is to split the network into a trainable "controller" and an "external" variable-size memory. It caused an outbreak of other neural network architectures with external memories (see Sec. 2).

However, one aspect which has been usually neglected so far is the efficiency of the memory access. Most of the proposed memory architectures have the $\Theta(n)$ access complexity, where $n$ is the size of the memory. It means that, for instance, copying a sequence of length $n$ requires performing $\Theta(n^2)$ operations, which is clearly unsatisfactory.

### 1.1 Our contribution

We propose a novel memory module for neural networks, called Hierarchical Attentive Memory (HAM). The HAM module is generic and can be used as a building block of larger neural architectures. Its crucial property is that it scales well with the memory size — the memory access requires only $\Theta(\log n)$ operations, where $n$ is the size of the memory. This complexity is achieved by using a new attention mechanism based on a binary tree with leaves corresponding to memory cells. The novel attention mechanism is not only faster than the standard one used in Deep Learning (Bahdanau et al., 2014), but it also facilities learning algorithms due to a built-in bias towards operating on intervals.

---

[*]Equal contribution.

We show that an LSTM augmented with HAM is able to learn algorithms for tasks like merging, sorting or binary searching. In particular, it is the first neural network, which we are aware of, that is able to learn to sort from pure input-output examples and generalizes well to input sequences much longer than the ones seen during the training. Moreover, the learned sorting algorithm runs in time $\Theta(n \log n)$. We also show that the HAM memory itself is capable of simulating different classic memory structures: a stack, a FIFO queue and a priority queue.

## 2 RELATED WORK

In this section we mention a number of recently proposed neural architectures with an external memory, which size is independent of the number of the model parameters.

**Memory architectures based on attention**    Attention is a recent but already extremely successful technique in Deep Learning. This mechanism allows networks to *attend* to parts of the (potentially preprocessed) input sequence (Bahdanau et al., 2014) while generating the output sequence. It is implemented by giving the network as an auxiliary input a linear combination of input symbols, where the weights of this linear combination can be controlled by the network. Attention mechanism was used to access the memory in Neural Turing Machines (NTMs) proposed by Graves et al. (2014). It was the first paper, that explicitly attempted to train a computationally universal neural network and achieved encouraging results.

The Memory Network (Weston et al., 2014) is an early model that attempted to explicitly separate the memory from computation in a neural network model. The followup work of (Sukhbaatar et al., 2015) combined the memory network with the soft attention mechanism, which allowed it to be trained with less supervision. In contrast to NTMs, the memory in these models is non-writeable. Another model without writeable memory is the Pointer Network (Vinyals et al., 2015), which is very similar to the attention model of Bahdanau et al. (2014). Despite not having a memory, this model was able to solve a number of difficult algorithmic problems, like the Convex Hull and the approximate 2D TSP.

All of the architectures mentioned so far use standard attention mechanisms to access the memory and therefore memory access complexity scales linearly with the memory size.

**Memory architectures based on data structures**    Stack-Augmented Recurrent Neural Network (Joulin & Mikolov, 2015) is a neural architecture combining an RNN and a differentiable stack. Grefenstette et al. (2015) consider extending an LSTM with a stack, a FIFO queue or a double-ended queue and show some promising results. The advantage of the latter model is that the presented data structures have a constant access time.

**Memory architectures based on pointers**    In two recent papers (Zaremba & Sutskever, 2015; Zaremba et al., 2015) authors consider extending neural networks with nondifferentiable memories based on pointers and trained using Reinforcement Learning. The big advantage of these models is that they allow a constant time memory access. They were however only successful on relatively simple tasks.

Another model, which use a pointer-based memory and learns sub-procedures is the Neural Programmer-Interpreter (Reed & de Freitas, 2015). Unfortunately, it requires strong supervision in the form of execution traces. Different type of pointer-based memory was presented in Neural Random-Access Machine (Kurach et al., 2015), which is a neural architecture mimicking classic computers.

**Parallel memory architectures**    There are two recent memory architectures, which are especially suited for parallel computation. Grid-LSTM (Kalchbrenner et al., 2015) is an extension of LSTM to multiple dimensions. Another recent model of this type is Neural GPU (Kaiser & Sutskever, 2015), which can learn to multiply long binary numbers.

# 3 HIERARCHICAL ATTENTIVE MEMORY

In this section we describe our novel memory module called Hierarchical Attentive Memory (HAM). The HAM module is generic and can be used as a building block of larger neural network architectures. For instance, it can be added to feedforward or LSTM networks to extend their capabilities. To make our description more concrete we will consider a model consisting of an LSTM "controller" extended with a HAM module.

The high-level idea behind the HAM module is as follows. The memory is structured as a full binary tree with the leaves containing the data stored in the memory. The inner nodes contain some auxiliary data, which allows us to efficiently perform some types of "queries" on the memory. In order to access the memory, one starts from the root of the tree and performs a top-down descent in the tree, which is similar to the hierarchical softmax procedure (Morin & Bengio, 2005). At every node of the tree, one decides to go left or right based on the auxiliary data stored in this node and a "query". Details are provided in the rest of this section.

## 3.1 NOTATION

The model takes as input a sequence $x_1, x_2, \ldots$ and outputs a sequence $y_1, y_2, \ldots$. We assume that each element of these sequences is a binary vector of size $b \in \mathbb{N}$, i.e. $x_i, y_i \in \{0, 1\}^b$. Suppose for a moment that we only want to process input sequences of length $\leq n$, where $n \in \mathbb{N}$ is a power of two (we show later how to process sequences of an arbitrary length). The model is based on the full binary tree with $n$ leaves. Let $V$ denote the set of the nodes in that tree (notice that $|V| = 2n - 1$) and let $L \subset V$ denote the set of its leaves. Let $l(e)$ for $e \in V \setminus L$ be the left child of the node $e$ and let $r(e)$ be its right child. We will now present the inference procedure for the model and then discuss how to train it.

## 3.2 INFERENCE

The high-level view of the model execution is presented in Fig. 1. The hidden state of the model consists of two components: the hidden state of the LSTM controller (denoted $h_{\text{LSTM}} \in \mathbb{R}^l$ for some $l \in \mathbb{N}$) and the hidden values stored in the nodes of the HAM tree. More precisely, for every node $e \in V$ there is a hidden value $h_e \in \mathbb{R}^d$. These values change during the recurrent execution of the model, but we drop all timestep indices to simplify the notation.

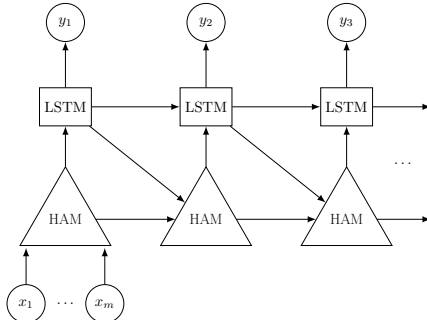

Figure 1: The LSTM+HAM model consists of an LSTM controller and a HAM module. The execution of the model starts with the initialization of HAM using the *whole* input sequence $x_1, x_2, \ldots, x_m$. At each timestep, the HAM module produces an input for the LSTM, which then produces an output symbol $y_t$. Afterwards, the hidden states of the LSTM and HAM are updated.

The parameters of the model describe the input-output behaviour of the LSTM, as well as the following 4 transformations, which describe the HAM module: EMBED : $\mathbb{R}^b \to \mathbb{R}^d$, JOIN : $\mathbb{R}^d \times \mathbb{R}^d \to \mathbb{R}^d$, SEARCH : $\mathbb{R}^d \times \mathbb{R}^l \to [0, 1]$ and WRITE : $\mathbb{R}^d \times \mathbb{R}^l \to \mathbb{R}^d$. These transformations may be represented by arbitrary function approximators, e.g. Multilayer Perceptrons (MLPs). Their meaning will be described soon.

The details of the model are presented in 4 figures. Fig. 2a describes the initialization of the model. Each recurrent timestep of the model consists of three phases: the *attention* phase described in Fig. 2b, the *output* phase described in Fig. 2c and the *update* phase described in Fig. 2d. The whole timestep can be performed in time $\Theta(\log n)$.

The HAM parameters describe only the 4 mentioned transformations and hence the number of the model parameters does not depend on the size of the binary tree used. Thus, we can use the model to process the inputs of an arbitrary length by using big enough binary trees. It is not clear that the same set of parameters will give good results across different tree sizes, but we showed experimentally that it is indeed the case (see Sec. 4 for more details).

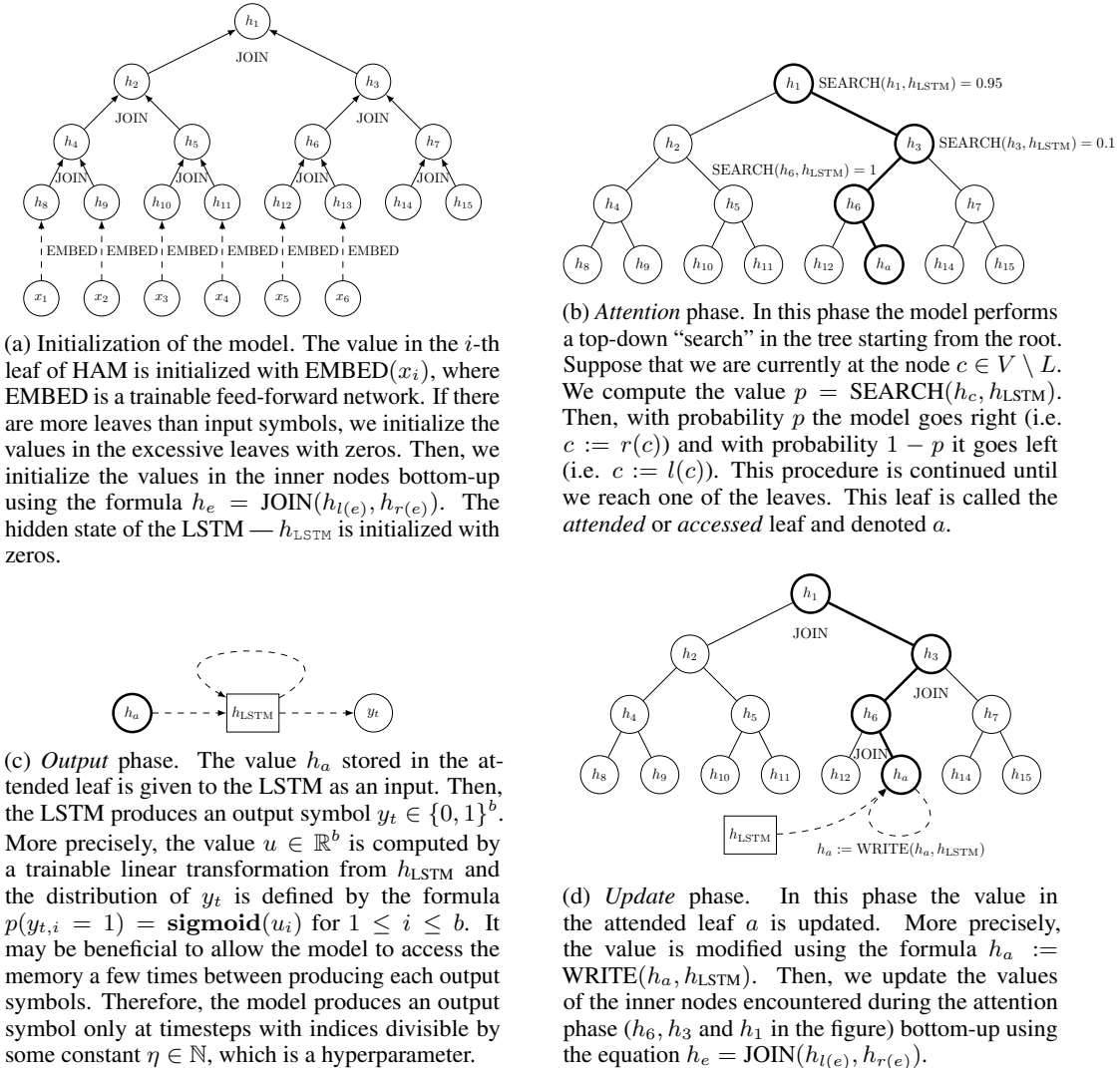

(a) Initialization of the model. The value in the $i$-th leaf of HAM is initialized with $\text{EMBED}(x_i)$, where EMBED is a trainable feed-forward network. If there are more leaves than input symbols, we initialize the values in the excessive leaves with zeros. Then, we initialize the values in the inner nodes bottom-up using the formula $h_e = \text{JOIN}(h_{l(e)}, h_{r(e)})$. The hidden state of the LSTM — $h_{\text{LSTM}}$ is initialized with zeros.

(b) *Attention* phase. In this phase the model performs a top-down "search" in the tree starting from the root. Suppose that we are currently at the node $c \in V \setminus L$. We compute the value $p = \text{SEARCH}(h_c, h_{\text{LSTM}})$. Then, with probability $p$ the model goes right (i.e. $c := r(c)$) and with probability $1 - p$ it goes left (i.e. $c := l(c)$). This procedure is continued until we reach one of the leaves. This leaf is called the *attended* or *accessed* leaf and denoted $a$.

(c) *Output* phase. The value $h_a$ stored in the attended leaf is given to the LSTM as an input. Then, the LSTM produces an output symbol $y_t \in \{0,1\}^b$. More precisely, the value $u \in \mathbb{R}^b$ is computed by a trainable linear transformation from $h_{\text{LSTM}}$ and the distribution of $y_t$ is defined by the formula $p(y_{t,i} = 1) = \textbf{sigmoid}(u_i)$ for $1 \le i \le b$. It may be beneficial to allow the model to access the memory a few times between producing each output symbols. Therefore, the model produces an output symbol only at timesteps with indices divisible by some constant $\eta \in \mathbb{N}$, which is a hyperparameter.

(d) *Update* phase. In this phase the value in the attended leaf $a$ is updated. More precisely, the value is modified using the formula $h_a := \text{WRITE}(h_a, h_{\text{LSTM}})$. Then, we update the values of the inner nodes encountered during the attention phase ($h_6$, $h_3$ and $h_1$ in the figure) bottom-up using the equation $h_e = \text{JOIN}(h_{l(e)}, h_{r(e)})$.

Figure 2: The model. One timestep consists of three phases presented in Figures (b)–(d).

We decided to represent the transformations defining HAM with MLPs with ReLU (Nair & Hinton, 2010) activation function in all neurons except the output layer of SEARCH, which uses sigmoid activation function to ensure that the output may be interpreted as a probability. Moreover, the network for WRITE is enhanced in a similar way as Highway Networks (Srivastava et al., 2015), i.e. $\text{WRITE}(h_a, h_{\text{LSTM}}) = T(h_a, h_{\text{LSTM}}) \cdot H(h_a, h_{\text{LSTM}}) + (1 - T(h_a, h_{\text{LSTM}})) \cdot h_a$, where $H$ and $T$ are two MLPs with sigmoid activation function in the output layer. This allows the WRITE transformation to easily leave the value $h_a$ unchanged.

## 3.3 TRAINING

In this section we describe how to train our model from purely input-output examples using REIN-FORCE (Williams, 1992). In Appendix B we also present a different variant of HAM which is fully differentiable and can be trained using end-to-end backpropagation.

Let $x, y$ be an input-output pair. Recall that both $x$ and $y$ are sequences. Moreover, let $\theta$ denote the parameters of the model and let $A$ denote the sequence of all decisions whetherto go left or right made during *the whole execution* of the model. We would like to maximize the log-probability of producing the correct output, i.e.

$$\mathcal{L} = \log p(y|x, \theta) = \log \left( \sum_A p(A|x, \theta) p(y|A, x, \theta) \right).$$

This sum is intractable, so instead of minimizing it directly, we minimize a variational lower bound on it:

$$\mathcal{F} = \sum_A p(A|x, \theta) \log p(y|A, x, \theta) \leq \mathcal{L}.$$

This sum is also intractable, so we approximate its gradient using the REINFORCE, which we briefly explain below. Using the identity $\nabla p(A|x, \theta) = p(A|x, \theta) \nabla \log p(A|x, \theta)$, the gradient of the lower bound with respect to the model parameters can be rewritten as:

$$\nabla \mathcal{F} = \sum_A p(A|x, \theta) \Big[ \nabla \log p(y|A, x, \theta) + \log p(y|A, x, \theta) \nabla \log p(A|x, \theta) \Big] \qquad (1)$$

We estimate this value using Monte Carlo approximation. For every $x$ we sample $\widetilde{A}$ from $p(A|x, \theta)$ and approximate the gradient for the input $x$ as $\nabla \log p(y|\widetilde{A}, x, \theta) + \log p(y|\widetilde{A}, x, \theta) \nabla \log p(\widetilde{A}|x, \theta)$. Notice that this gradient estimate can be computed using normal backpropagation if we substitute the gradients in the nodes[1] which sample whether we should go left or right during the *attention* phase by

$$\underbrace{\log p(y|\widetilde{A}, x, \theta)}_{\text{return}} \nabla \log p(\widetilde{A}|x, \theta).$$

This term is called REINFORCE gradient estimate and the left factor is called a *return* in Reinforcement Learning literature. This gradient estimator is unbiased, but it often has a high variance. Therefore, we employ two standard variance-reduction technique for REINFORCE: *discounted returns* and *baselines* (Williams, 1992). Discounted returns means that our return at the $t$-th timestep has the form $\sum_{t \leq i} \gamma^{i-t} \log p(y_i|\widetilde{A}, x, \theta)$ for some discount constant $\gamma \in [0, 1]$, which is a hyperparameter. This biases the estimator if $\gamma < 1$, but it often decreases its variance.

For the lack of space we do not describe the *baselines* technique. We only mention that our baseline is case and timestep dependent: it is computed using a learnable linear transformation from $h_{\text{LSTM}}$ and trained using MSE loss function. The whole model is trained with the Adam (Kingma & Ba, 2014) algorithm. We also employ the following three training techniques:

**Different reward function**  During our experiments we noticed that better results may be obtained by using a different reward function for REINFORCE. More precisely, instead of the log-probability of producing the correct output, we use the percentage of the output bits, which have the probability of being predicted correctly (given $\widetilde{A}$) greater than 50%, i.e. our discounted return is equal $\sum_{t \leq i, 1 \leq j \leq b} \gamma^{i-t} \Big[ p(y_{i,j}|\widetilde{A}, x, \theta) > 0.5 \Big]$. Notice that it corresponds to the Hamming distance between the most probable outcome accordingly to the model (given $\widehat{A}$) and the correct output.

**Entropy bonus term**  We add a special term to the cost function which encourages exploration. More precisely, for each sampling node we add to the cost function the term $\frac{\alpha}{H(p)}$, where $H(p)$ is the entropy of the distribution of the decision, whether to go left or right in this node and $\alpha$ is an exponentially decaying coefficient. This term goes to infinity, whenever the entropy goes to zero, what ensures some level of exploration. We noticed that this term works better in our experiments than the standard term of the form $-\alpha H(p)$ (Williams, 1992).

**Curriculum schedule**  We start with training on inputs with lengths sampled uniformly from $[1, n]$ for some $n = 2^k$ and the binary tree with $n$ leaves. Whenever the error drops below some threshold, we increment the value $k$ and start using the bigger tree with $2n$ leaves and inputs with lengths sampled uniformly from $[1, 2n]$.

---

[1] For a general discussion of computing gradients in computation graphs, which contain stochastic nodes see (Schulman et al., 2015).

## 4 EXPERIMENTS

In this section, we evaluate two variants of using the HAM module. The first one is the model described in Sec. 3, which combines an LSTM controller with a HAM module (denoted by LSTM+HAM). Then, in Sec. 4.3 we investigate the "raw" HAM (without the LSTM controller) to check its capability of acting as classic data structures: a stack, a FIFO queue and a priority queue. It would be also interesting to get some insight into the algorithms learned by the model. In Appendix A we present an example execution on the `Sort` task.

### 4.1 TEST SETUP

For each test that we perform, we apply the following procedure. First, we train the model with memory of size up to $n = 32$ using the curriculum schedule described in Sec. 3.3. The model is trained using the minibatch Adam algorithm with exponentially decaying learning rate. We use random search to determine the best hyper-parameters for the model. We use gradient clipping (Pascanu et al., 2012) with constant 5. The depth of our MLPs is either 1 or 2, the LSTM controller has $l = 20$ memory cells and the hidden values in the tree have dimensionality $d = 20$. Constant $\eta$ determining a number of memory accesses between producing each output symbols (Fig. 2c) is equal either 1 or 2. We always train for 100 epochs, each consisting of 1000 batches of size 50. After each epoch we evaluate the model on 200 validation batches without learning. When the training is finished, we select the model parameters that gave the lowest error rate on validation batches and report the error using these parameters on fresh $2, 500$ random examples.

We report two types of errors: a test error and a generalization error. The test error shows how well the model is able to fit the data distribution and generalize to unknown cases, assuming that cases of similar lengths were shown during the training. It is computed using the HAM memory with $n = 32$ leaves, as the percentage of output *sequences*, which were predicted incorrectly. The lengths of test examples are sampled uniformly from the range $[1, n]$. Notice that we mark the whole output sequence as incorrect even if only one bit was predicted incorrectly, e.g. a hypothetical model predicting each bit incorrectly with probability $1\%$ (and independently of the errors on the other bits) has an error rate of $96\%$ on *whole sequences* if outputs consist of 320 bits.

The generalization error shows how well the model performs with enlarged memory on examples with lengths exceeding $n$. We test our model with memory 4 times bigger than the training one. The lengths of input sequences are now sampled uniformly from the range $[2n + 1, 4n]$.

During testing we make our model fully deterministic by using the most probable outcomes instead of stochastic sampling. More precisely, we assume that during the *attention phase* the model decides to go right iff $p > 0.5$ (Fig. 2b). Moreover, the output symbols (Fig. 2c) are computed by rounding to zero or one instead of sampling.

### 4.2 LSTM+HAM

We evaluate the model on a number of algorithmic tasks described below:

1. `Reverse`: Given a sequence of 10-bit vectors, output them in the reversed order., i.e. $y_i = x_{m+1-i}$ for $1 \le i \le m$, where $m$ is the length of the input sequence.

2. `Search`: Given a sequence of pairs $x_i = \textbf{key}_i || \textbf{value}_i$ for $1 \le i \le m - 1$ sorted by keys and a query $x_m = q$, find the smallest $i$ such that $\textbf{key}_i = q$ and output $y_1 = \textbf{value}_i$. Keys and values are 5-bit vectors and keys are compared lexicographically. The LSTM+HAM model is given only two timesteps ($\eta = 2$) to solve this problem, which forces it to use a form of binary search.

3. `Merge`: Given two *sorted* sequences of pairs — $(p_1, v_1), \ldots, (p_m, v_m)$ and $(p'_1, v'_1), \ldots, (p'_{m'}, v'_{m'})$, where $p_i, p'_i \in [0, 1]$ and $v_i, v'_i \in \{0, 1\}^5$, merge them. Pairs are compared accordingly to their priorities, i.e. values $p_i$ and $p'_i$. Priorities are unique and sampled uniformly from the set $\{\frac{1}{300}, \ldots, \frac{300}{300}\}$, because neural networks cannot easily distinguish two real numbers which are very close to each other. Input is encoded as

$x_i = p_i || v_i$ for $1 \le i \le m$ and $x_{m+i} = p_i' || v_i'$ for $1 \le i \le m'$. The output consists of the vectors $v_i$ and $v_i'$ sorted accordingly to their priorities[2].

4. `Sort`: Given a sequence of pairs $x_i = \textbf{key}_i || \textbf{value}_i$ sort them in a stable way[3] accordingly to the lexicographic order of the keys. Keys and values are 5-bit vectors.

5. `Add`: Given two numbers represented in binary, compute their sum. The input is represented as $a_1, \ldots, a_m, \textbf{+}, b_1, \ldots, b_m, \textbf{=}$ (i.e. $x_1 = a_1, x_2 = a_2$ and so on), where $a_1, \ldots, a_m$ and $b_1, \ldots, b_m$ are bits of the input numbers and $\textbf{+}, \textbf{=}$ are some special symbols. Input and output numbers are encoded starting from the *least* significant bits.

Every example output shown during the training is finished by a special "End Of Output" symbol, which the model learns to predict. It forces the model to learn not only the output symbols, but also the length of the correct output.

We compare our model with 2 strong baseline models: encoder-decoder LSTM (Sutskever et al., 2014) and encoder-decoder LSTM with attention (Bahdanau et al., 2014), denoted LSTM+A. The number of the LSTM cells in the baselines was chosen in such a way, that they have more parameters than the biggest of our models. We also use random search to select an optimal learning rate and some other parameters for the baselines and train them using the same curriculum scheme as LSTM+HAM.

The results are presented in Table 1. Not only, does LSTM+HAM solve all the problems almost perfectly, but it also generalizes very well to much longer inputs on all problems except `Add`. Recall that for the generalization tests we used a HAM memory of a different size than the ones used during the training, what shows that HAM generalizes very well to new sizes of the binary tree. We find this fact quite interesting, because it means that parameters learned from a small neural network (i.e. HAM based on a tree with 32 leaves) can be successfully used in a different, bigger network (i.e. HAM with 128 memory cells).

In comparison, the LSTM with attention does not learn to merge, nor sort. It also completely fails to generalize to longer examples, which shows that LSTM+A learns rather some statistical dependencies between inputs and outputs than the real algorithms.

The LSTM+HAM model makes a few errors when testing on longer outputs than the ones encountered during the training. Notice however, that we show in the table the percentage of output sequences, which contain *at least one* incorrect bit. For instance, LSTM+HAM on the problem `Merge` predicts incorrectly only $0.03\%$ of output bits, which corresponds to $2.48\%$ of incorrect output sequences. We believe that these rare mistakes could be avoided if one trained the model longer and chose carefully the learning rate schedule. One more way to boost generalization would be to simultaneously train the models with different memory sizes and shared parameters. We have not tried this as the generalization properties of the model were already very good.

Table 1: Experimental results. The upper table presents the error rates on inputs of the same lengths as the ones used during training. The lower table shows the error rates on input sequences 2 to 4 times longer than the ones encountered during training. LSTM+A denotes an LSTM with the standard attention mechanism. Each error rate is a percentage of *output sequences*, which contained at least one incorrectly predicted bit.

| test error | LSTM | LSTM+A | LSTM+HAM |
|---|---|---|---|
| Reverse | 73% | 0% | **0%** |
| Search | 62% | 0.04% | **0.12%** |
| Merge | 88% | 16% | **0%** |
| Sort | 99% | 25% | **0.04%** |
| Add | 39% | 0% | **0%** |
| **2-4x longer inputs** | LSTM | LSTM+A | **LSTM+HAM** |
| Reverse | 100% | 100% | **0%** |
| Search | 89% | 0.52% | **1.68%** |
| Merge | 100% | 100% | **2.48%** |
| Sort | 100% | 100% | **0.24%** |
| Add | 100% | 100% | **100%** |
| **Complexity** | $\Theta(1)$ | $\Theta(n)$ | $\Theta(\textbf{log n})$ |

---

[2] Notice that we earlier assumed for the sake of simplicity that the input sequences consist of *binary* vectors and in this task the priorities are *real* values. It does not however require any change of our model. We decided to use real priorities in this task in order to diversify our set of problems.

[3] Stability means that pairs with equal keys should be ordered accordingly to their order in the input sequence.

## 4.3 RAW HAM

In this section, we evaluate "raw" HAM module (without the LSTM controller) to see if it can act as a drop-in replacement for 3 classic data structures: a stack, a FIFO queue and a priority queue. For each task, the network is given a sequence of PUSH and POP operations in an *online* manner: at timestep $t$ the network sees only the $t$-th operation to perform $x_t$. This is a more realistic scenario for data structures usage as it prevents the network from cheating by peeking into the future.

Raw HAM module differs from the LSTM+HAM model from Sec. 3 in the following way:

- The HAM memory is initialized with zeros.
- The $t$-th output symbol $y_t$ is computed using an MLP from the value in the accessed leaf $h_a$.
- Notice that in the LSTM+HAM model, $h_{\text{LSTM}}$ acted as a kind of "query" or "command" guiding the behaviour of HAM. We will now use the values $x_t$ instead. Therefore, at the $t$-th timestep we use $x_t$ instead of $h_{\text{LSTM}}$ whenever $h_{\text{LSTM}}$ was used in the original model, e.g. during the *attention* phase (Fig. 2b) we use $p = \text{SEARCH}(h_c, x_t)$ instead of $p = \text{SEARCH}(h_c, h_{\text{LSTM}})$.

We evaluate raw HAM on the following tasks:

1. `Stack`: The "PUSH $x$" operation places the element $x$ (a 5-bit vector) on top of the stack, and the "POP" returns the last added element and removes it from the stack.

2. `Queue`: The "PUSH $x$" operation places the element $x$ (a 5-bit vector) at the end of the queue and the "POP" returns the oldest element and removes it from the queue.

3. `PriorityQueue`: The "PUSH $x$ $p$" operations adds the element $x$ with priority $p$ to the queue. The "POP" operation returns the value with the highest priority and remove it from the queue. Both $x$ and $p$ are represented as 5-bit vectors and priorities are compared lexicographically. To avoid ties we assume that all elements have different priorities.

Model was trained with the memory of size up to $n = 32$ with operation sequences of length $n$. Sequences of PUSH/POP actions for training were selected randomly. The $t$-th operation out of $n$ operations in the sequence was POP with probability $\frac{t}{n}$ and PUSH otherwise. To test generalization, we report the error rates with the memory of size $4n$ on sequences of operations of length $4n$.

The results presented in Table 2 show that HAM simulates a stack and a queue perfectly with no errors whatsoever even for memory 4 times bigger. For the `PriorityQueue` task, the model generalizes almost perfectly to large memory, with errors only in $0.2\%$ of output sequences.

Table 2: Results of experiments with the raw version of HAM (without the LSTM controller). Error rates are measured as a percentage of operation sequences in which *at least one* POP query was not answered correctly.

| Task | Test Error | Generalization Error |
|---|---|---|
| `Stack` | 0% | 0% |
| `Queue` | 0% | 0% |
| `Priority Queue` | 0.08% | 0.2% |

## 5 COMPARISON TO OTHER MODELS

As far as we know, our model is the first one which is able to learn a sorting algorithm from pure input-output examples. Although this problem was considered in the original NTM paper, the error rate achieved by the NTM is in fact quite high – the log-likelihood of the correct output was equal around 20 bits on outputs consisting of 128 bits. In comparison our model learns to solve almost perfectly - only $0.04\%$ of the outputs produced by our model contain at least one incorrect bit.

Reed & de Freitas (2015) shown that an LSTM is able to learn to sort short sequences, but it fails to generalize to inputs longer than the ones seen during the training. It is quite clear that an LSTM cannot learn a "real" sorting algorithm, because it uses a bounded memory independent of the length of the input. The Neural Programmer-Interpreter (Reed & de Freitas, 2015) is a neural network architecture, which is able to learn bubble sort, but it requires strong supervision in the form of

execution traces. In comparison, our model can be trained from pure input-output examples, which is crucial if we want to use it to solve problems for which we do not know any algorithms.

An important feature of neural memories is their efficiency. Our HAM module in comparison to many other recently proposed solutions is effective and allows to access the memory in $\Theta(\log(n))$ complexity. In the context of learning algorithms it may sound surprising that among all the architectures mentioned in Sec. 2 the only ones, which can copy a sequence of length $n$ without $\Theta(n^2)$ operations are: Reinforcement-Learning NTM (Zaremba & Sutskever, 2015), the model from (Zaremba et al., 2015), Neural Random-Access Machine (Kurach et al., 2015), and Queue-Augmented LSTM (Grefenstette et al., 2015). However, the first three models have been only successful on relatively simple tasks. The last model was successful on some synthetic tasks from the domain of Natural Language Processing, which are very different from the tasks we tested our model on, so we cannot directly compare the two models.

## 6 CONCLUSIONS

We presented a new memory architecture for neural networks called Hierarchical Attentive Memory. Its crucial property is that it scales well with the memory size — the memory access requires only $\Theta(\log n)$ operations. This complexity is achieved using a new attention mechanism based on a binary tree. The model proved to be successful on a number of algorithmic problems. The future work is to apply this or similar architecture to very long real-world sequential data like books or DNA sequences.

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

# A    EXAMPLE: HAM SORTING

We present some insights into the algorithms learned by the LSTM+HAM model, by investigating the hidden representations $h_e$ learned for a variant of the problem `Sort` in which we sort 4-bit vectors lexicographically[4]. For demonstration purposes, we use a small tree with $n = 8$ leaves and each node's hidden state has size $d = 6$ values.

The trained network performs sorting perfectly. It attends to the leaves in the order corresponding to the order of the sorted input values, i.e. at every timestep HAM attends to the leaf corresponding to the smallest input value among the leaves, which have not been attended so far.

It would be interesting to exactly understand the algorithm used by the network to perform this operation. A natural solution to this problem would be to store in each hidden node $e$ the smallest input value among the (unattended so far) leaves *below* $e$ together with the information whether the smallest value is in the right or the left subtree under $e$.

In the Fig. 3 we present two timesteps of our model. The LSTM controller is not presented to simplify the exposition. The input sequence is presented on the left, below the tree: $x_1 = $ `0000`, $x_2 = $ `1110`, $x_3 = $ `1101` and so on. The 2x3 grids in the nodes of the tree represent the values $h_e \in \mathbb{R}^6$. White cells correspond to value 0 and non-white cells correspond to values $> 0$.

The lower-rightmost cells are presented in pink, because we managed to decipher the meaning of this coordinate for the inner nodes. This coordinate in the node $e$ denotes whether the minimum in the subtree (among the values unattended so far) is in the right or left subtree of $e$. Value greater than 0 (pink in the picture) means that the minimum is in the right subtree and therefore we should go right while visiting this node in the *attention* phase.

In the first timestep the leftmost leaf (corresponding to the input `0000`) is accessed. Notice that the last coordinates (shown in pink) are updated appropriately, e.g. the smallest unattended value at the beginning of the second timestep is `0101`, which corresponds to the 6-th leaf. It is in the right subtree under the root and accordingly the last coordinate in the hidden value stored in the root is high (i.e. pink in the figure).

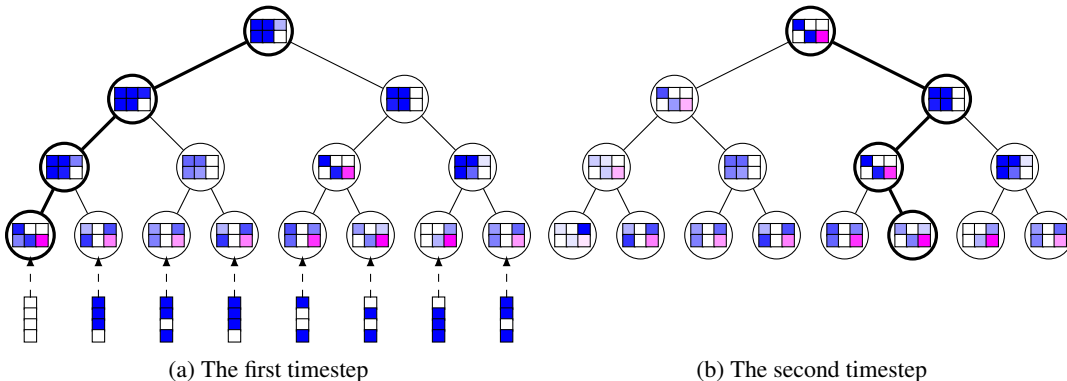

(a) The first timestep (b) The second timestep

Figure 3: An exemplary input sequence and the state of HAM after initialization (left) and after first timestep (right).

---

[4] In the problem `Sort` considered in the experimental results, there are separate keys and values, which forces the model to learn stable sorting. Here, for the sake of simplicity, we consider the simplified version of the problem and do not use separate keys and values.

## B  USING SOFT ATTENTION

One of the open questions in the area of designing neural networks with attention mechanisms is whether to use a *soft* or *hard* attention. The model described in the paper belongs to the latter class of attention mechanisms as it makes hard, stochastic choices. The other solution would be to use a soft, differentiable mechanism, which attends to a linear combination of the potential attention targets and do not involve any sampling. The main advantage of such models is that their gradients can be computed exactly.

We now describe how to modify the model to make it fully differentiable ("DHAM"). Recall that in the original model the leaf which is attended at every timestep is sampled stochastically. Instead of that, we will now at every timestep compute for every leaf $e$ the probability $p(e)$ that this leaf would be attended if we used the stochastic procedure described in Fig. 2b. The value $p(e)$ can be computed by multiplying the probabilities of going in the right direction from all the nodes on the path from the root to $e$.

As the input for the LSTM we then use the value $\sum_{e \in L} p(e) \cdot h_e$. During the *write* phase, we update the values of *all* the leaves using the formula $h_e := p(e) \cdot \mathrm{WRITE}(h_e, h_{\mathrm{ROOT}}) + (1 - p(e)) \cdot h_e$. Then, in the *update* phase we update the values of *all* the inner nodes, so that the equation $h_e = \mathrm{JOIN}(h_{l(e)}, h_{r(e)})$ is satisfied for each inner node $e$. Notice that one timestep of the soft version of the model takes time $\Theta(n)$ as we have to update the values of all the nodes in the tree. Our model may be seen as a special case of Gated Graph Neural Network (Li et al., 2015).

This version of the model is fully differentiable and therefore it can be trained using end-to-end backpropagation on the log-probability of producing the correct output. We observed that training DHAM is slightly easier than the REINFORCE version. However, DHAM does not generalize as well as HAM to larger memory sizes.

