# Peer review of "Learning Efficient Algorithms with Hierarchical Attentive Memory"

_ICLR 2017 — rejected_

[Official Review · AnonReviewer3 · rating 5 · confidence 4 · 15 Dec 2016]
**interesting idea and model but not clear that it actually works for long sequences**

The authors introduce a new memory model which allows memory access in O(log n) time.

Pros:
* The paper is well written and everything is clear.
* It's a new model and I'm not aware of a similar model.
* It's clear that memory access time is an issue for longer sequences and it is clear how this model solves this problem.

Cons:
* The motivation for O(log n) access time is to be able to use the model on very long sequences. While it is clear from the definition that the computation time is low because of its design, it is not clear that the model will really generalize well to very long sequences.
* The model was also not tested on any real-world task.

I think such experiments should be added to show whether the model really works on long sequences and real-world tasks, otherwise it is not clear if this is a useful model.

[Official Review · AnonReviewer2 · rating 5 · confidence 5 · 16 Dec 2016]
**interesting idea, weak experiments**

This paper introduces a novel hierarchical memory architecture for neural networks, based on a binary tree with leaves corresponding to memory cells.  This allows for O(log n) memory access, and experiments additionally demonstrate ability to solve more challenging tasks such as sorting from pure input-output examples and dealing with longer sequences.

The idea of the paper is novel and well-presented, and the memory structure seems reasonable to have advantages in practice. However, the main weakness of the paper is the experiments. There is no experimental comparison with other external memory-based approaches (e.g. those discussed in Related Work), or experimental analysis of computational efficiency given overhead costs (beyond just computational complexity) despite that being one of the main advantages. Furthermore, the experimental setups are relatively weak, all on artificial tasks with moderate increases in sequence length.  Improving on these would greatly strengthen the paper, as the core idea is interesting.

[Official Review · AnonReviewer1 · rating 3 · confidence 4 · 16 Dec 2016 (modified: 19 Dec 2016)]

This paper proposes to use a hierarchical softmax to speed up attention based memory addressing in memory augmented network (e.g. NTM, memNN…).

The model build a hierarchical softmax on top of the input sequence then at each time step SEARCH for the most relevant input to predict the next output (this search is discrete), and use its corresponding embedding to update the state of an LSTM that will then produce the output. Finally the embedding of the used input is update by a WRITE function (an LSTM working that takes hidden state of the other LSTM as an input). The model has a discrete component (the SEARCH) and is thus trained with REINFORCE. In the experimental section they test their approach on several algorithmic tasks such as search, sort...

The main advantage of replacing the full softmax by a hierarchical softmax is that during inference, the complexity goes from O(N) to O(log(N)). It would be great to see if the gain in complexity allows to tackle problem which are a few orders of magnitude bigger than the one addressed with full softmax. However the authors only test on toy sequences up to 32 tokens, which is quite small. 

The model requires a relatively complex search mechanism that can only be trained with REINFORCE. While this seems to work on problems with relatively small and simple sequences, it would be great to see how performance changes with the size of the problem. 

Overall, while the idea of replacing the softmax in the attention mechanism by a hierachical softmax is appealing, this work is not quite convincing yet. Their approach is not very natural, may be hard to train and may not be that simple to scale. The experiment section is very weak.

[Final Decision · Program Chairs · 06 Feb 2017]
**ICLR committee final decision**

All three reviewers point to significant deficiencies. No response or engagement from the authors (for the reviews). I see no basis for supporting this paper.